# Anomaly Detection using Normalizing Flows and Contrastive Data

## Abstract

Detecting test data deviating from training data is a central problem for safe and robust machine learning. Likelihoods learned by a generative model, e.g., a normalizing flow via standard log-likelihood training, perform poorly as an anomaly score. We propose to use an unlabelled auxiliary dataset and a probabilistic outlier score for anomaly detection. We use a self-supervised feature extractor trained on the auxiliary dataset and train a normalizing flow on the extracted features by maximizing the likelihood on in-distribution data and minimizing the likelihood on the auxiliary dataset. We show that this is equivalent to learning the normalized positive difference between the in-distribution and the auxiliary feature density. We conduct experiments on benchmark datasets and show a robust improvement compared to likelihood, likelihood ratio methods and state-of-the-art anomaly detection methods.

## 1 Introduction

The performance of neural nets is governed by the availability of vast amount of data, and - with enough training data - neural nets can achieve superhuman performance in various tasks such as classification. If an image at test time is not similar to the training images, but stems from another distribution, classical neural networks may fail to classify the image (Heaven et al., 2019; Boult et al., 2019). The detection of such anomalies, also called out-of-distribution detection or outlier detection, is a central problem in modern machine learning and is crucial for safe and trustable neural networks: At test time, we want to know if a given input stems from the same distribution as during training time, in order to know if we can trust our trained net on that input.

This may not significantly influence the prediction performance at test time, either because the events are that rare, or because the test data is similar to the training data, but introduces a major security risk: If the camera of a self-driving car has a malfunction or something is occluding the view, the car should be able to detect the rareness of the situation and should not use the input of the camera. The use of anomaly detection to improve performance for rare events or unseen data is manifold: One can use anomaly detection while training to purify the data set (Zhao et al., 2019b), at train time to give rare training data points a stronger training signal (Steininger et al., 2021), or at test time to find and react to anomalies (Wang et al., 2021). In contrast to discriminative networks, where the texture (Geirhos et al., 2020) or the background (Beery et al., 2018) can be enough to get a sufficient performance, for anomaly detection the network must distinguish between the in-distribution data and all other -unknown- data. The network therefore must "understand" what the in-distribution data characterizes and following Richard Feynman's famous saying "What I cannot create, I do not understand", we believe that generative models are therefore the most promising approach to anomaly detection. We turn to the fast growing field of normalizing flows (Kobyzev et al., 2021), which allow exact density estimation, fast sampling, and suffer (almost) no mode collapse. While an important line of research deals with detecting small anomalies within an image, so-called defects, mostly in an industrial setting (e.g., Roth et al. (2021a)), we focus in this work on semantic single-shot anomaly detection. The goal is to find anomalies on image level: An anomaly is an image of a different semantic class, which is not known at training time. We train e.g. solely on images of the class deer of the CIFAR-10 dataset and want to distinguish between images of the in-distribution class deer and the other CIFAR-10 classes at test

time. In contrast to distribution-based anomaly detection where a set of many test-samples/measurements is compared to the training distribution (see e.g. Coluccia et al. (2013); Bouyeddou et al. (2018)) we focus on the single-shot setting, where the inlier vs. outlier decision must be made independently for individual data points. Inspired by the work of Ren et al. (2019) and Schirrmeister et al. (2020), who used the ratio of likelihoods for anomaly detection, we modify the training objective for normalizing flows to learn the positive difference of two distributions: An inlier distribution $p$ and an auxiliary distribution $q$. We prove that this is achieved by maximizing the log-likelihood on data which stems from $p$ while minimizing the log-likelihood on data stemming from $q$. We use the negative log-likelihood under this usually intractable difference distribution given by our normalizing flow as anomaly score and show that this new score is especially suited for anomaly detection. We argue and demonstrate experimentally that the difference $p - q$ is much more robust than the ratio $p/q$: In regions where $p$ and $q$ are small, there are only few training instances, and the estimators of $p$ and $q$ will accordingly be very noisy. The quotient between two noisy small numbers can become very big, whereas their difference remains small. We use IMAGENET (Deng et al., 2009) as auxiliary distribution, a popular dataset of natural images. We do not use label information, but see the auxiliary dataset as a collection of unlabeled images. We motivate the use of such an auxiliary dataset twofold: Collecting unlabeled images is easy and cheap, but labeling them is expensive and introduces a potential bias in the training. We find that using the difference between the two distributions results in an improved anomaly score. Since applying normalizing flows on images is computationally intensive, we do not train directly on images but employ a dimensionality reducing feature extractor trained self-supervised on IMAGENET. This fits the ongoing development of employing pretrained models to use prior knowledge (Bergman et al., 2020). An schematic overview of our method is shown in figure 2. To give an introductory example, we compare the anomaly scores for all CIFAR10 classes given by a standard normalizing flow, the ratio method and our proposed method trained on the deer class of the CIFAR10 dataset as in-distribution data and IMAGENET as contrastive dataset in figure 1. One can see that our method separates semantic distant classes better compared to existing methods.

We focus on normalizing flows as density estimators, since they are universal approximators working well for high dimensional data. However, the new objective and theory is applicable to any density estimator trainable with explicit maximum likelihood, as is required for our new objective in equation 3. This includes Mixture-of-Gaussian density estimators or binning methods in lower dimensional settings.

Our method "Anomaly Detection using Normalizing Flows and Contrastive Data" combines a generative model with exact density estimation (normalizing flow) and a new training objective for density estimators. We employ a pre-trained feature extractor for further performance improvement. The novelties of our approach are the following:

- Developing a new likelihood-based anomaly score with state-of-the-art performance on benchmark datasets

- Presenting a new objective to train density estimators

- Proving the equivalence of our new objective to a negative log-likelihood training of an intractable difference distribution better suited for anomaly detection

- Experiments using Normalizing Flows that show theoretical and practical advantages compared to the likelihood-ratio method

## 2 Related work

**Anomaly detection** Anomaly detection, also known as out-of-distribution or novelty detection, is the task of detecting unknown images at test time. See Salehi et al. (2021) for a extensive discussion of the field. In this work, we will focus on semantic anomaly detection, by either using one class of a given dataset as inliers and all other classes as anomalies (one-vs-rest setting) or by using a complete dataset as inliers and comparing against other datasets (dataset-vs-dataset setting). Most related work can be categorized as either reconstruction based or using a representation ansatz: In a reconstruction approach, the model tries to reconstruct a given image, and the score relies on the difference between the reconstruction and the

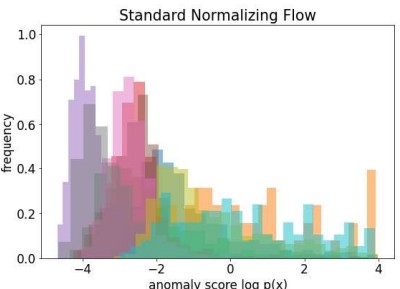 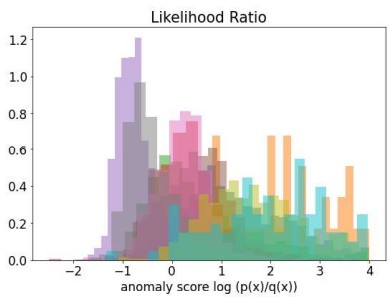 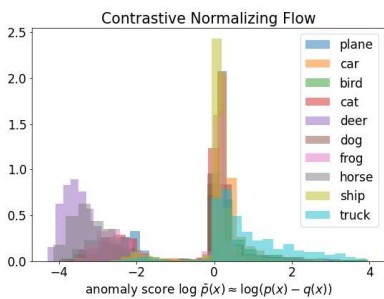

Figure 1: Histogram of the anomaly scores of all CIFAR-10 classes under models trained with the inlier class deer. The semantically similar classes deer and horse are difficult to separate for all methods, our method "contrastive normalizing flow" achieves good separation for the other classes.

original image. This idea goes back to Japkowicz et al. (1995) and has been applied to various domains, e.g. time series (Zhang et al., 2020), medical diagnosis (Lu and Xu, 2018) and flight data (Memarzadeh et al., 2020). Recent models used for image data are reconstruction with a memory module in the latent space (Gong et al., 2019) or combinations of VAE and GAN (Perera et al., 2019). An interesting new idea is the combination of VAE and energy-based models by Yoon et al. (2021), where the reconstruction loss is interpreted as the energy of the model. For the representation approach the model tries to learn a feature space representation and introduces a measure for the outlierness in this representation: Ruff et al. (2018) map all data inside a hypersphere and define the score as the distance to the center. Another approach uses transformations to either define negatives for contrastive learning (Tack et al., 2020) or to directly train a classifier (Bergman and Hoshen, 2020). Zong et al. (2018) fit a gaussian mixture model to the latent space of an autoencoder.

**Normalizing flows**   While most generative models are either able to easily generate new samples like GANs or VAEs or to give an exact (possible unnormalized) density estimation (like energy-based models), Normalizing flows, first introduced by Rezende and Mohamed (2015) are able to perform both tasks.Normalizing flows rely on the change of variable formula to compute the likelihood of the the data and therefore need a tractable Jacobian: Most normalizing flows achieve this by either using mathematical "tricks" (e.g., Rezende and Mohamed (2015)), as autoregressive models which restrict the Jacobian to a triangular shape (e.g., Kingma et al. (2016)) or by special architectures which allow invertibility (e.g., RealNVP by Dinh et al. (2017)). RealNVP makes use of coupling blocks to apply an affine transformation on a subset of the features conditioned on the subsets complement.

**Density of normalizing flows for anomaly detection**   The use of generative models seems like a perfect fit for anomaly detection. Classifiers tend to focus on features to distinguish the given classes, whereas generative models need to include all relevant information to be able to generate new samples. Methods with exact density estimation (e.g., normalizing flows, energy-based methods) directly offer a good anomaly score via the density. Unfortunately, normalizing flows work poorly in the case of anomaly detection when employed directly on images: Various works showed that the likelihood is dominated by low-level statistics (Nalisnick et al., 2019b) and pixel-correlations (Kirichenko et al., 2020) and fail to detect anomalies (see also Zhang et al. (2021)). There have been multiple attempts to improve the anomaly score directly by introducing a complexity measure (Serrà et al., 2020), using typicality (Nalisnick et al., 2019a), using hierarchies (Schirrmeister et al., 2020) or employing ensembles (Choi et al., 2019).

**Likelihood ratio for anomaly detection**   Another line of research is using the ratio of two likelihood models as anomaly score: Ren et al. (2019) use an augmented version of the in-distribution as auxiliary distribution and computed the likelihood ratio with Pixel-CNN, an autoregressive generative model. Schirrmeister et al. (2020) train separate normalizing flows on the in-distribution data and Tiny Images as auxiliary dataset. They require the in-distribution dataset to be included in the auxiliary dataset to meet their hierarchical principle.

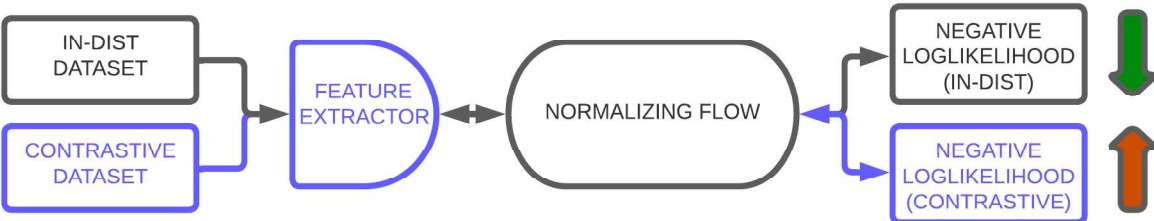

Figure 2: Schematic overview of our method. Standard normalizing flow in black. The feature extractor is an optional extension, which further improves the performance for image data. We do not use an feature extrator for tabular data and for our toy experiments.

**Feature extractor**  With the broad availability of trained deep models and their good generalization capacity, the use of such models as feature extractor has gained popularity: Most implementations use the output of intermediate layers of a model trained as a discriminator on an auxiliary dataset (e.g. IMAGENET by Deng et al. (2009)) as features: Prominent examples are ResNet (He et al., 2016), used by e.g., Cohen and Hoshen (2021) and Roth et al. (2021b), or ViT (Dosovitskiy et al., 2021), used by Cohen and Avidan (2021) for anomaly detection. We argue that the use of label information introduces a bias, especially when working on similar datasets to IMAGENET as CIFAR-10 (Krizhevsky et al.) or CIFAR-100) and propose to use a feature extractor trained in a self-supervised fashion via a contrastive learning objective (Chen et al., 2020). Their and the improved method by He et al. (2020) showed remarkable results on down-stream tasks without relying on label information.

**Combining feature extractors and normalizing flows**  An existing line of work uses the representations given by pretrained feature extractors to detect outliers: Cohen and Hoshen (2021) use the intermediate layer of a pretrained ResNet as the feature representation and use the sum of distances to the k-nearest Neighbours as their anomaly score. Yu et al. (2021) and Rudolph et al. (2021) train a unconditioned normalizing flow on a feature space of a pretrained feature extractor trained on IMAGENET, while Gudovskiy et al. (2021) use conditional normalizing flows (Ardizzone et al., 2019). All of these methods focus mainly on defect detection and localization, while our paper works on the task of semantic anomaly or out-of distribution detection. In contrast to MOCO (He et al., 2020), which is used in this work for the feature extraction, all their feature extractors are trained in a supervised fashion.

## 3  Background

### 3.1  Normalizing flows

Normalizing flows are a class of generative models, which allow exact density estimation and fast sampling. A comprehensive guide to normalizing flows can be found in Papamakarios et al. (2021). A normalizing flow maps the given data via an invertible transformation $T^{-1}$ to a normal distribution (of the same dimensionality) by minimizing the empirical KL-Divergence between the transformed data distribution in the latent space and a multivariate normal distribution. This is equivalent to minimizing the negative log-likelihood of the training data under the model. This likelihood can be calculated by the change of variable formula. Therefore, the Jacobian of the transformation needs to be tractable:

$$L(\theta) = -\sum_{\boldsymbol{x} \in X} \log p_\theta(\boldsymbol{x}) = \sum_{\boldsymbol{x} \in X} \frac{||T^{-1}(\boldsymbol{x})||_2^2}{2} - \log |\mathrm{Jac}_T(\boldsymbol{x})|, \tag{1}$$

where $X$ is the training data, $\theta$ are the parameters of the invertible transformation $T$ and $\mathrm{Jac}_T$ denotes the determinant of the Jacobian. To apply the change of variable formula in equation 1 the transformation needs

to be invertible and needs a tractable determinant of the Jacobian. We achieve this by using the realNVP architecture established by Dinh et al. (2017).

### 3.2 Feature extractor

To achieve good anomaly detection performance, the feature extractor should be trained on diverse images and without class information. We use a pretrained MOCO (He et al., 2020) feature extractor, which is trained self-supervised on IMAGENET. Self-supervised contrastive methods learn a representation by maximizing the similarity of different views of the same image in the feature space. For a training step, every training image is augmented twice, by augmentations consisting of color distortions, horizontal flipping, and random cropping. These augmented images are feed into an encoder network and for a positive pair $\boldsymbol{z}_0, \boldsymbol{z}_1$ and negative representations $\boldsymbol{z}_2, ..., \boldsymbol{z}_N$ (other augmented images) the following loss is optimized for all augmented images:

$$l_0 = -\log \frac{\exp(sim(\boldsymbol{z}_0, \boldsymbol{z}_1))/\tau}{\sum_{k=1}^{N} \exp(sim(\boldsymbol{z}_0, \boldsymbol{z}_i)/\tau}.$$ (2)

$\tau$ is a temperature scalar and $sim(\boldsymbol{x}, \boldsymbol{y})$ denotes the cosine similarity (cosine of the angle) between $\boldsymbol{x}$ and $\boldsymbol{y}$. The contrastive loss is trained without any label information. We also run an ablation experiment with inceptionV3 as alternative feature extractor and report the results in the appendix in A.3.

## 4 Method: Contrastive normalizing flow

We introduce the contrastive normalizing flow as a novel way to train normalizing flows: We adapt the training of normalizing flows by maximizing the log-likelihood on in-distribution data $\boldsymbol{x} \sim p$ and minimizing the log-likelihood on a broader auxiliary dataset $\boldsymbol{y} \sim q$. The intuition behind this approach is simple: We want the model to learn high likelihoods on in-distribution data and low likelihoods everywhere else. We use the negative log likelihood given by the model trained with this novel training objective as anomaly score. We train our model with the objective in equation 3, for illustrative purposes we rearrange the objective into equation 6, which gives a better intuition what this objective achieves:

$$L = \min_{\theta} \mathbb{E}_{\boldsymbol{x} \sim p}[-\log p_\theta(\boldsymbol{x})] - \mathbb{E}_{\boldsymbol{y} \sim q}[-\log p_\theta(\boldsymbol{y})]$$ (3)

$$= \min_{\theta} - \int \log p_\theta(\boldsymbol{x})[p(\boldsymbol{x}) - q(\boldsymbol{x})]d\boldsymbol{x}$$ (4)

$$= \min_{\theta} - \int_{\text{supp}(p>q)} \log p_\theta(\boldsymbol{x})[p(\boldsymbol{x}) - q(\boldsymbol{x})]d\boldsymbol{x} + \min_{\theta} \int_{\text{supp}(q>p)} \log p_\theta(\boldsymbol{x})[q(\boldsymbol{x}) - p(\boldsymbol{x})]d\boldsymbol{x}$$ (5)

$$= \frac{1}{C} \min_{\theta} \mathbb{E}_{\boldsymbol{x} \sim C(p-q)1_{\{p>q\}}}[-\log p_\theta(\boldsymbol{x})] + \min_{\theta} \int_{\text{supp}(q>p)} \log p_\theta(\boldsymbol{x})[q(\boldsymbol{x}) - p(\boldsymbol{x})]d\boldsymbol{x},$$ (6)

where for the optimal $\hat{\theta}$ the second term in equation 6 gives $p_{\hat{\theta}}(\boldsymbol{x}) \to 0 \; \forall \boldsymbol{x} \in \text{supp}(q > p)$. Because we are only interested in anomaly detection and therefore in a threshold on the anomaly score (the negative log density given by the model trained with the new objective), we clamp the negative log-likelihood of the contrastive dataset. Without clamping, our training objective in equation 3 is dominated by large negative log-likelihoods of the contrastive dataset in the second expectation value. We discuss this clamping in the experimental section 5.2.1. The first term in equation 6 is the negative log-likelihood objective for $\bar{p} = C(p-q)1_{\{p>q\}}$, with normalizing constant $C$. This term results in $p_{\hat{\theta}}(\boldsymbol{x}) = C(p-q) \; \forall \boldsymbol{x} \in \text{supp}(p > q)$,

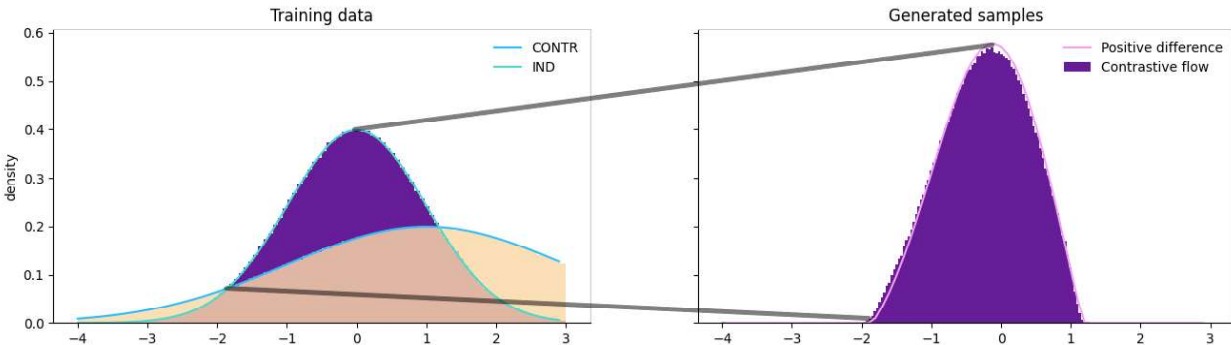

Figure 3: Example of an 1D contrastive normalizing flow: We train on samples $x \sim p = N(0, 1)$ as in-distribution and $y \sim q = N(1, 2)$ as contrastive dataset. The training distributions $p$ and $q$ are shown on the left. The learned distribution of our contrastive normalizing flow method and the analytic normalised positive difference between the ground truth densities are shown on the right. The black lines connect equivalent points in both diagrams.

as we show below. For the first term in equation 6 holds:

$$L_1 = \frac{1}{C} \min_\theta \mathbb{E}_{\boldsymbol{x} \sim \bar{p}}[-\log p_\theta(\boldsymbol{x})] \tag{7}$$

$$= \frac{1}{C} \min_\theta \mathbb{E}_{\boldsymbol{x} \sim \bar{p}}[\log \frac{\bar{p}(\boldsymbol{x})}{p_\theta(\boldsymbol{x})} - \log \bar{p}(\boldsymbol{x})] \tag{8}$$

$$= \frac{1}{C} \min_\theta KL(\bar{p}, p_\theta) + H(\bar{p}), \tag{9}$$

where $KL(\bar{p}, p_\theta)$ is the KL-divergence between $\bar{p}$ and $p_\theta$ with the global minimum for $p_{\hat{\theta}} = \bar{p}$ and the second term is the entropy $H$ of $\bar{p}$ independent of $\theta$. By training our model with equation 3, we learn the normalized positive difference between in-distribution and auxiliary distribution. This is in contrast to a standard normalizing flow, where we learn the distribution of the in-distribution samples. We use the negative logarithm of this new - usually intractable - difference density as anomaly score. We give an 1D toy example of a contrastive normalizing flow in figure 3. The pseudocode for our training procedure is given in section 4.1. We argue that the difference distribution is especially suited for anomaly detection: When using a broader distribution as auxiliary contrastive distribution we can assume that $p(\boldsymbol{x}) > q(\boldsymbol{x})$ on in-distribution data. In regions where the contrastive distribution $q(\boldsymbol{x})$ has a high density the learned positive difference will be small or zero. This behaviour is similar to a likelihood ratio method of Ren et al. (2019). An advantage of our method compared to the likelihood ratio is that our learned density $p_\theta$ is well defined in areas where $p(\boldsymbol{x})$ and $q$ are very small, i.e., where the in-distribution data and the contrastive distribution data are sparse. The learned density $p_\theta$ is normalized and therefore integrates to 1. Thus, in the areas where both distributions $p$ and $q$ have no support it is zero or almost zero. When comparing $p_\theta(\boldsymbol{x})$ at test time to a threshold to see whether it is an inlier (bigger than threshold) or outlier (smaller than threshold) data points outside of the support of $p$ and $q$ will be reliably detected as outliers. In contrast, the density ratio $p(\boldsymbol{x})/q(\boldsymbol{x})$ is ill-defined in regions where both $p(\boldsymbol{x})$ and $q(\boldsymbol{x})$ have little or no support. The division of two small numbers can either be very large or very small. This leads to problems as the experiments in section 5 show, because the data which one might want to detect as outlier can be very far from the contrastive distribution which was used during training − a outlier can be anything, and here the likelihood ratio fails. We show in a 2D toy experiment in section 5.1, that the likelihood ratio has the highest inlier score far away from the inlier and contrastive distribution while our model is able to capture the in-distribution well. We also show this behaviour in a real world setting in section 5.3.2.

When separating the probability density into a semantic part and a low-level pixel correlation part, these common low-level correlations cancel out since they are the same for in-distribution and out-of-distribution (Ren et al., 2019). We argue that this is even more beneficial for our difference distribution: We separate the

feature dimensions of our data in a high level semantic representation $s$ and a low-level feature representation $f$, where the low level pixel feature conditional distribution $p(f|s)$ is the same for all natural images. This strong assumption of Ren et al. (2019) is not necessary for our theory to hold but is purely there to help explain the method and give the reader some intuition. We can now rewrite the difference between the in-distribution $p$ and the auxiliary distribution $q$ as

$$p(\boldsymbol{x}) - q(\boldsymbol{x}) = p(\boldsymbol{f}|\boldsymbol{s})p(\boldsymbol{s}) - q(\boldsymbol{f}|\boldsymbol{s})q(\boldsymbol{s}) = p(\boldsymbol{f}|\boldsymbol{s})(p(\boldsymbol{s}) - q(\boldsymbol{s})).$$

This results in a low score when either the low-level features have a low density given the semantic content of the image or the semantic likelihood of the image is higher for the broader auxiliary distribution than for the inlier distribution. We find both cases important classes of anomalies. The anomaly score is the negative log density given by our model, which is equivalent to the negative log-likelihood of the positive difference density described in equation 6. By setting a threshold $\delta$, all inputs with a anomaly score $> \delta$ are classified as anomalies. For evaluating, we use area under the receiver operating characteristic (AUROC).

### 4.1 Training Pseudocode

To clarify how the model is trained, we report the training process in pseudocode in algorithm 1.

---
**Algorithm 1** Training process
---
With MOCO feature extractor, density estimator $T_\theta$ and $\mathrm{nll}_\theta(\boldsymbol{x}) = -\log \mathrm{probability}_{T_\theta}(\boldsymbol{x})$
**for all** epochs **do**
    **for all** batches $\boldsymbol{x} \sim$ inlier data and $\boldsymbol{y} \sim$ contrastive data **do**
        **for all** $\boldsymbol{x}_i \sim \boldsymbol{x}$ and $\boldsymbol{y}_j \sim \boldsymbol{y}$ **do**
            $\hat{\boldsymbol{x}}_i = \mathrm{MOCO}(\boldsymbol{x}_i), \hat{\boldsymbol{y}}_j = \mathrm{MOCO}(\boldsymbol{y}_j)$
            $\mathrm{lossPos}_i = \mathrm{nll}_\theta(\hat{\boldsymbol{x}}_i)$
            $\mathrm{lossNeg}_j = \mathrm{nll}_\theta(\hat{\boldsymbol{y}}_j)$
            $\mathrm{lossNeg}_i = \mathrm{clamp}(\mathrm{lossNeg}_i, \mathrm{None}, \mathrm{tsh})$
        **end for**
        $\mathrm{loss} = \frac{1}{N}\sum_i^N \mathrm{lossPos}_i - \frac{1}{M}\sum_j^M \mathrm{lossNeg}_j$
        $\mathrm{GradientStep}(\theta, \mathrm{loss})$
    **end for**
**end for**

---

### 4.2 Choice of contrastive distribution

We investigate the influence of the contrastive distribution in experimental section 5 and discuss the results and give recommendations on the choice in subsection 5.5.

## 5 Experiments

**Experimental outline** To show the advantages of our method we conduct several experiments: First we show on low dimensional toy data that the experiments agree with the derived theory and demonstrate in 2D that our contrastive normalizing flow succeeds where the ratio approach of Ren et al. (2019) struggles. Then we show in real life examples using IMAGENET as contrastive dataset that our model shows good anomaly performance for various in-distribution data using the Cifar-10 and Cifar-100 classes. We also show that the method works if the in-distribution dataset and the contrastive dataset are dissimilar using CelebA and IMAGENET. We do ablation studies to investigate the role of the contrastive distribution: We show the change of performance for contaminating the contrastive data with inlier data, for different mixtures between the broad IMAGENET distribution and a narrow CIFAR10 class as in-distribution and in an additional experiment for informed anomaly detection, where we have a-priori knowledge on the outliers.

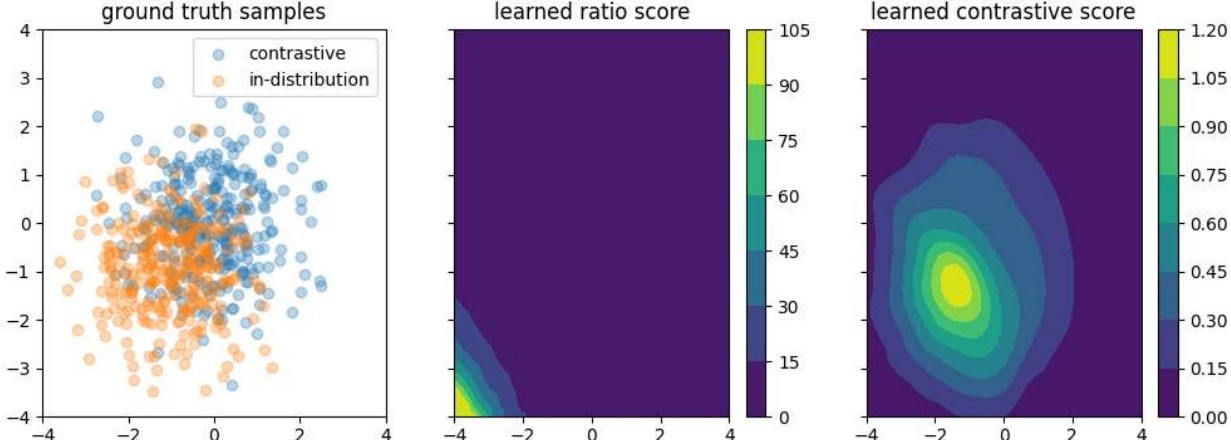

Figure 4: Learned in-distribution scores for the density ratio method and our contrastive normalizing flow on a 2D toy examples. For better visualisation, we plot the exponential of the in-distribution score (equivalent to the exponential of the negative anomaly score used throughout the paper). On the left the ground truth samples for the in-distribution and the contrastive data are shown. While the ratio method has the highest in-distribution scores far away from both distributions in the lower left, the contrastive flow in-distribution score captures the inliers well.

## 5.1 Toy experiments

**1D Toy Example** We conduct a qualitative 1D experiment to demonstrate that our derived theory holds by training a contrastive normalizing flow on samples $x \sim p = N(0, 1)$ as in-distribution and $y \sim q = N(1, 2)$ as contrastive dataset. We show in figure 3, that the density learned by the model exactly matches the positive difference density as described in equation 6. We also do an ablation study on the role of the threshold-hyperparameter $\epsilon$ in the appendix A.3.

**2D Toy Example** To show the theoretical advantage of the learned difference density over the density ratio method, we conduct a 2D toy experiment: We learn a contrastive normalizing flow on a normal distribution centered at [1,1] as in-distribution and a normal distribution centered on the origin as contrastive distribution. For the ratio method, we train a separate normalizing flow on each distribution, and take the ratio at inference time. As we see in figure 4, our contrastive normalizing captures the in-distribution data, while the ratio method has the highest in-distribution scores far from both distributions. This is not surprising, as the analytic log ratio of the two Gaussian distributions goes to infinity for $x \to [-\infty, -\infty]$.

## 5.2 Real world data

### 5.2.1 Implementation details

To extract useful features, we use a MOCO encoder pretrained on IMAGENET (He et al., 2020). As features we use the output of the network, this is in contrast to the original MOCO implementation, where classification is done via an MLP head on the penultimate layer. For our case, working with the last layer results in a better performance for anomaly detection. Because the contrastive MOCO objective is invariant under changes of the norm of the features, we normalize all features to a hypersphere and add small noise afterwards to obtain a valid density. For the normalizing flow implementation, we use the "Framework for Easily Invertible Architectures" (Ardizzone et al., 2018-2022). Unless otherwise specified, we use eight of their "AllinOne"-blocks for our architecture. We list all hyperparameter for training in section A.1 in the appendix.

**Contrastive Loss Clamping** For our loss, we apply a small deviation from the theory: The second term in equation 6 results in the theoretical optimum $p_\theta(\boldsymbol{x}) = 0 \; \forall \boldsymbol{x} \in \text{supp}(q > p)$, but the loss diverges because

$\lim_{x\to 0} \log x$ is unbounded. To handle this mismatch, we clamp $\log p(\boldsymbol{x})$ at a threshold $\epsilon$ as a lower bound on the likelihood. Therefore the objective leads to $\log p(\boldsymbol{x}) < \epsilon \ \forall \boldsymbol{x} \in \text{supp}(q > p)$, which we find to be sufficient for successful anomaly detection. For our experiments, we found $\epsilon = 0$ sufficient for a good anomaly detection performance. For unknown datasets, we recommend to train a standard normalizing flow first and using the distribution of the in-distribution likelihoods to determine a constant clamping parameter, e.g. the 10% quantile of the in-distribution log density plus a fixed offset (e.g., $\ln(10) = 2.3$) . We investigate the influence of the threshold $\epsilon$ in the appendix in A.3 and find that the anomaly detection performance is constant under reasonable changes of $\epsilon$.

**MOCO Finetuning** For datasets with image sizes significantly smaller than the images used in the MOCO implementation with 224 by 224 pixels, the features are dominated by upsampling artefacts and are highly correlated between different images. To reduce this problem without changing the setup, we finetuned a MOCO trained on 244*244 images on smaller images by shrinking IMAGENET images to 32 by 32 pixels and then using upsamling to 224 by 224 pixels again. We discuss this in appendix A.3.

### 5.2.2 Benchmark methods

We used the PyDO library (Zhao et al., 2019a) to compare our results to standard anomaly detection techniques. We present results for PCA, KDE, and KNN conducted on the MOCO feature space, and show additional results for other methods and training directly on the image data (without feature extractor) in the appendix (see A.2). As a simple baseline, we use the mean squared error (MSE) to the mean of the feature space representations of the training set as an outlier score. This is equivalent to fitting a normal distribution to the feature space around the mean of the training data. This simple baseline already gives good results with the finetuned MOCO feature extractor. We extend the MSE to also employ the auxiliary distribution by taking the difference of the MSE to the mean of the inlier set and the contrastive set as anomaly score. This corresponds to the likelihood ratio of two normal distributions. We call this method MSE-ratio. As a third method (Flow), we train an unconditioned flow with the same architecture as our model on the MOCO feature representation and use the negative log-likelihood under the learned distribution as an outlier measure. We show that this results on datasets similar to the auxiliary distribution in a meaningful anomaly measure because of the semantic content of the MOCO features. As fourth baseline method (Flow-ratio), we train an additional flow (with the same architecture as our model) on the auxiliary dataset and use the likelihood ratio of the learned inlier distribution and the learned auxiliary distribution using the hierarchies of distribution method of Schirrmeister et al. (2020). In contrast to their work, the normalizing flows are trained on the MOCO feature space and not on the images directly. We also employ the outlier exposure method of Hendrycks et al. (2019a) on our architecture by training a standard normalizing flow on the inlier data and finetune afterwards with their likelihood margin loss using the auxiliary dataset. At last, we compare our method to "CSI: Novelty Detection via Contrastive Learning on Distributionally Shifted Instances" (Tack et al., 2020). This method is the unsupervised state of the art method for one-class anomaly detection on CIFAR-10. They train their model via contrastive learning, with transformed (shifted) instances of an image itself as additional negatives.

### 5.2.3 One-vs-rest - results

We run experiments on CIFAR-10, CIFAR-100 (Krizhevsky et al.) and celebA (Liu et al., 2015). We work in the one-vs-rest setting, where one class is used as inlier data, and all the other classes of the same dataset are used as anomalies at test time. For CIFAR-100, we show results for superclasses. For celebA, we divide the dataset into two classes by the given gender attribute and use one of the classes as inlier and the other one as anomalies.

**CIFAR-10 and CIFAR-100 superclasses** In the following experiment we want to investigate the performance for a broader contrastive distribution, which encloses the in-distribution: We evaluated on the CIFAR-10 and CIFAR-100 datasets as inlier distribution with the IMAGENET dataset as contrastive distribution. Note that all CIFAR-10 and -100 classes are a subset of the IMAGENET dataset. The discussion in Section 4 holds also in practice: The broader contrastive IMAGENET distribution with overlap of the CIFAR inlier distributions does not negatively affect the performance, in contrary our method achieves state of the art results on almost all classes when evaluated under a One-vs-Rest setting. We show qualitative

Table 1: AUROC scores on CIFAR-10 classes for the One-Vs-Rest setting. PCA, KDE and KNN are taken from the PyOD library (Zhao et al., 2019a). GOAD from Bergman and Hoshen (2020), CSI from Tack et al. (2020), and Rot+T(rans) from Hendrycks et al. (2019b). MSE, MSE-ratio, and Flow are three ablation methods of our method contrastive normalizing flow (CF). Flow-ratio is the 'Hierarchies of Distributions'-method by Schirrmeister et al. (2020), but applied on the MOCO feature space. OE denotes the outlier exposure method by Hendrycks et al. (2019a). All methods within one standard deviation of the best AUROC score per class are highlighted. † denotes methods using an auxiliary dataset.

| method | plane | car | bird | cat | deer | |
|---|---|---|---|---|---|---|
| **KDE** | 94.4 | **99.1** | 90.4 | 90.4 | 93.3 | |
| **PCA** | 94.2 | 98.9 | 90.6 | 90.4 | 93.2 | |
| **KNN** | 96.0 | 98.7 | 92.3 | 90.4 | 93.6 | |
| **GOAD** | 75.5 | 94.1 | 81.8 | 72.0 | 83.7 | |
| **Rot+T** | 77.5 | 96.9 | 87.3 | 80.9 | 92.7 | |
| **CSI** | 89.9 | **99.1** | 93.1 | 86.4 | **93.9** | |
| **MSE** | 94.6 | **99.1** | 90.4 | 90.5 | 93.7 | |
| **MSE-ratio** † | 92.8 | 98.5 | 89.8 | 89.7 | 91.8 | |
| **Flow** | 96.1 ± 0.2 | 97.5 ± 0.2 | 92.6 ± 0.2 | 89.8 ± 0.4 | 93.3 ± 0.3 | |
| **Flow-ratio** † | 95.9 ± 0.2 | 97.7 ± 0.4 | 93.5 ± 0.5 | 90.0 ± 0.1 | 93.2 ± 0.5 | |
| **OE**† | **96.5 ± 0.8** | **99.2 ± 0.1** | **92.9 ± 1.7** | **92.6 ± 0.6** | **93.8 ± 0.3** | |
| **CF (ours)**† | **96.9 ± 0.3** | **99.0 ± 0.1** | **94.6 ± 0.1** | **92.8 ± 0.4** | **93.5 ± 0.4** | |
| method | dog | frog | horse | ship | truck | mean |
| **KDE** | 92.1 | 96.2 | 94.7 | 98.5 | 98.2 | 94.7 |
| **PCA** | 93.0 | 96.6 | 95.0 | 98.6 | 98.3 | 94.9 |
| **KNN** | **95.7** | 98.0 | 95.8 | 98.5 | 97.3 | 95.6 |
| **GOAD** | 84.4 | 82.9 | 93.9 | 92.9 | 89.5 | 85.1 |
| **Rot+T** | 90.2 | 90.9 | 96.5 | 95.2 | 93.3 | 90.1 |
| **CSI** | 93.2 | 95.1 | **98.7** | 97.9 | 95.5 | 94.3 |
| **MSE** | 91.4 | 96.3 | 95.2 | **98.7** | 98.2 | 94.8 |
| **MSE-ratio** † | 92.5 | 95.4 | 94.3 | 98.3 | 97.6 | 94.1 |
| **Flow** | **95.7 ± 0.4** | **98.0 ± 0.4** | 94.7 ± 0.4 | 97.8 ± 0.0 | 96.6 ± 0.1 | 95.2 ± 0.2 |
| **Flow-ratio** † | **95.9 ± 0.1** | **98.2 ± 0.0** | 95.2 ± 0.3 | 97.5 ± 0.4 | 96.6 ± 0.5 | 95.4 ± 0.2 |
| **OE**† | 93.8 ±0.3 | 97.6 ±0.4 | 96.6 ± 0.5 | 98.4 ± 0.1 | **98.6 ± 0.3** | 96.0 ± 0.2 |
| **CF (ours)**† | **96.1 ± 0.4** | **98.2 ± 0.0** | 96.3 ± 0.2 | **98.6 ± 0.1** | **98.5 ± 0.0** | **96.5 ± 0.1** |

results in figure 5: The OOD images with the lowest anomaly score are also for the human eye close to the IN-DIST, while the IN-DIST images with the highest anomaly score are sensible outliers. We show quantitative results in table 1: One can see that our method beats the ablation methods reliably, while the flow-ratio ablation method performs similarly to the simple flow approach. We think this is a result of the training on the intermediate MOCO feature space. To further investigate our method, we show the confusion matrix for all classes in table 2: By looking at the failure cases, we can verify that the model focuses on semantic features: the two worst pairs are deer-horse and cat-dog, which are semantic similar classes. Truck and car have nearly perfect scores; only truck versus car fails (AUROC of about 90). This is also shown in figure 1, where we show the scores of all CIFAR-10 images for a model trained on the deer class as IN-DIST. We conduct the same experiments on the twenty CIFAR-100 superclasses and verify the CIFAR-10 results: Our method outperforms or is on par with the benchmarks and the baseline methods and shows state of the art anomaly detection performance. The CIFAR100 results for all superclasses and methods can be found in table 12 in the appendix.

**CelebA** In the following experiment we want to measure the performance for a contrastive distribution dissimilar to the in-distribution data: We evaluate our model on the CelebA dataset. We treat each class -we use the gender attribute given in the dataset- once as inlier distribution and the other class as test-time outlier distribution. Even though our contrastive distribution for training the contrastive normalizing flow and the flow-ratio method is the IMAGENET dataset, which does not contain close-up pictures of human

Table 2: Confusion matrix on CIFAR-10 for the contrastive normalizing flow. Every row shows results for a model trained on one Cifar-10 class as inlier distribution and evaluated against all other CIFAR-10 classes. Hard cases (AUROC<90) are printed bold.

|       | plane | car  | bird | cat  | deer | dog  | frog | horse | ship | truck | mean |
|-------|-------|------|------|------|------|------|------|-------|------|-------|------|
| plane |       | 98.3 | 96.5 | 98.7 | 98.2 | 99.4 | 98.1 | 98.2  | **88.3** | 97.2 | 96.9 |
| car   | 99.6  |      | 99.9 | 99.8 | 99.9 | 99.9 | 99.8 | 99.8  | 99.2 | 93.0  | 99.0 |
| bird  | 94.9  | 99.9 |      | 94.6 | **83.5** | 97.3 | **88.8** | 93.3  | 99.3 | 99.8  | 94.6 |
| cat   | 97.9  | 99.7 | 93.9 |      | **88.7** | **72.3** | 90.9 | 92.0  | 99.1 | 99.7  | 92.8 |
| deer  | 97.7  | 99.4 | 92.3 | 93.8 |      | 96.1 | 93.2 | **70.5** | 98.8 | 99.7  | 93.5 |
| dog   | 99.5  | 99.6 | 98.3 | **82.9** | 95.8 |      | 98.6 | 91.2  | 99.4 | 99.8  | 96.1 |
| frog  | 98.8  | 99.7 | 96.8 | 95.9 | 95.9 | 98.8 |      | 99.4  | 99.1 | 99.9  | 98.2 |
| horse | 98.6  | 99.8 | 97.1 | 96.4 | **82.3** | 94.4 | 99.0 |       | 99.4 | 99.8  | 96.3 |
| ship  | 93.8  | 98.3 | 99.6 | 99.4 | 99.5 | 99.6 | 99.5 | 99.5  |      | 98.4  | 98.6 |
| truck | 98.4  | 91.2 | 99.8 | 99.6 | 99.7 | 99.8 | 99.7 | 99.5  | 98.7 |       | 98.5 |

Table 3: AUROC scores of models trained on the celebA dataset split into the provided gender attribute and evaluated against the other class using contrastive normalizing, CSI, and our ablation methods. LOF, ABOD, MDC and KNN are taken from the PyOD library (Zhao et al., 2019a). All models within one standard deviation of the best performing model are highlighted. We neglect the small standard deviation of the deterministic methods introduced by the MOCO preprocessing. Please note that the model only sees images of the in-distribution class and the unlabeled auxiliary IMAGENET dataset at training time.

| **IN-DIST** | LOF  | ABOD | MCD  | KNN  | CSI  | Flow | Ratio | OE | **CF (our)** |
|-------------|------|------|------|------|------|------|-------|----|--------------|
| female      | 52.5 | 68.7 | 69.7 | 74.1 | 68.3 | 75.8 ± 2.1 | 76.7 ± 1.8 | 80.3± 1.2 | **83.8 ± 3.2** |
| male        | 54.2 | 76.5 | 76.6 | 82.3 | 79.1 | **86.7 ± 0.8** | **86.4 ± 2.3** | **88.2 ± 1.4** | **87.1 ± 2.6** |
| average     | 53.4 | 72.6 | 73.2 | 78.2 | 73.7 | 81.3 ± 1.1 | 81.5 ± 1.5 | **84.3 ± 0.9** | **85.3 ± 2.1** |

faces, our method beats the baselines. However, the lacking overlap of the contrastive distribution with the test-time outlier distribution explains why the contrastive approaches such as the flow-ratio method and our contrastive flow do not outperform the other baselines by a clear margin and the overall performance leaves room for improvement. We report the results in table 3.

### 5.2.4 Dataset-vs-dataset

Next we investigate the setting of multi-modal in-distribution data and distant OOD samples. For the dataset-vs-dataset setting, the complete unlabeled dataset is used as the inlier distribution at training time. We train a contrastive normalizing flow on CIFAR-10 and use CIFAR-100, celebA and SVHN as anomaly distributions at test time. We report the AUROC scores in table 4. This setting is well studied, e.g., by Nalisnick et al. (2019b).

Table 4: AUROC scores in the dataset-vs-dataset setting. We train on CIFAR-10 as IN-DIST and IM-AGENET as auxiliary contrastive distribution. Additional results are reported in table 9. The standard deviations are taken over three runs. All models within one standard deviation of the best performing model are highlighted.

| method     | CIFAR-10 vs CIFAR-100 | CIFAR-10 vs SVHN | CIFAR-10 vs celebA | mean |
|------------|-----------------------|------------------|--------------------|------|
| MSE        | 70.0 ± 0.0            | 20.0 ± 0.0       | 92.2 ± 0.0         | 60.7 ± 0.0 |
| MSE-Ratio† | 74.8 ± 0.0            | 42.0 ± 0.0       | 91.5 ± 0.0         | 69.4 ± 0.0 |
| Flow       | 82.9± 0.6             | 65.4 ± 3.3       | **100.0 ± 0.1**    | 82.8 ± 1.1 |
| Flow-ratio | **84.7 ± 0.3**        | 68.3 ± 6.6       | **100 ± 0.0**      | 83.3 ± 2.2 |
| OE†        | 81.3 ± 0.9            | 65.0 ± 2.3       | **99.9 ± 0.1**     | 82.8 ± 0.8 |
| CF (ours)  | **84.4 ± 0.3**        | **90.3 ± 1.5**   | **99.8 ± 0.3**     | **91.5 ± 0.5** |

## 5.3 Investigation of the contrastive distribution

To investigate the role of the contrastive distribution and to get insights on the difference between our method and the ratio method proposed by Ren et al. (2019), we run ablation studies on different dataset mixtures as contrastive distribution. We run most of our ablation experiments on the deer and the horse class, as our model and the baselines show the worst performance on these two classes (see table 2 for the full confusion matrix) and experiments are most informative here.

### 5.3.1 In-distribution data in contrastive distribution

In this ablation experiment the performance for a contrastive distribution containing in-distribution samples is measured. When including inliers in the contrastive dataset, we expect to see no change in performance following our theoretical derivation: With in-distribution data $p$ and pure contrastive data $q$ (and learned positive difference $\bar{p} = \bar{C}(p - q)1_{p>q}$), we get for the mixed contrastive data $q_{new} = Dp + (1 - D)q$:

$$\bar{p}_{new} = C(p - q_{new})1_{p>q_{new}} \qquad (10)$$
$$= C(p - Dp - (1 - D)q)1_{p>Dp+(1-D)q} \qquad (11)$$
$$= C((1 - D)p - (1 - D)q)1_{(1-D)p>(1-D)q} \qquad (12)$$
$$= C(1 - D)(p - q)1_{p>q} \qquad (13)$$
$$= \bar{p}, \qquad (14)$$

with $C(1-D) = \bar{C}$. To show this qualitatively, we use the horse class of the CIFAR10 dataset as inlier class and a mixture of IMAGENET and in-distribution data as contrastive dataset and report the AUROC scores for different mixing factors in figure 6. One can see that the performance of our method stays constant for $D > 0$, where the ratio method drops in performance as $D$ approaches zero and the contrastive distribution contains more and more inliers. For $D = 0$, as in-distribution and contrastive distribution coincide, both models do not get a useful training or detection signal and anomaly detection fails for both (AUROC score approx. 50).

### 5.3.2 Narrow contrastive distribution

In this ablation study, we want to investigate the performance for a narrow contrastive dataset. We use the deer class of CIFAR10 as in-distribution data and a mixture of IMAGENET and the horse class of CIFAR10 as contrastive dataset. We plot in figure 7 the AUROC scores for the horse class included in the mixture and the other CIFAR10 classes. While the performance of the ratio method drops significantly (AUROC 96.5 to 75.6) when using a single narrow CIFAR10 class as contrastive dataset (mixture ratio equal to 0), our method only drops slightly (AUROC of 96.7 to 93). We plot in figure 8 the histograms for in-distribution, contrastive and out-of-distribution data for only the horse class as contrastive dataset. We see that the ratio method fails and assigns a lower anomaly score to out-of-distribution examples than to in-distribution data. We conclude that our method is more robust than the ratio method for outliers far from the inlier and contrastive distribution as discussed in section 4 and the results of the toy experiment in 5.1 transfer to real world data.

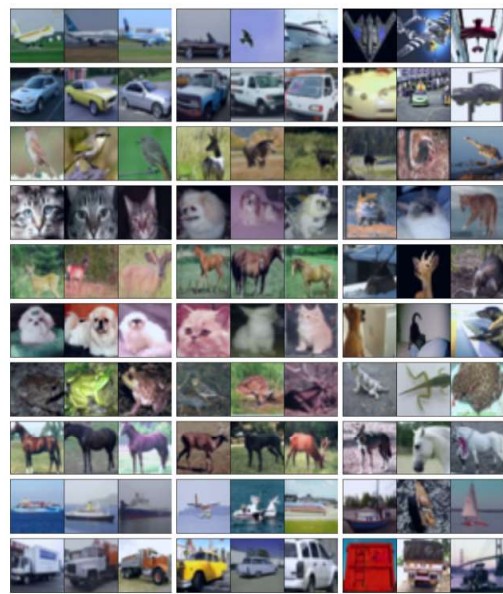

Figure 5: Visual examples from the IN-DIST and OOD test sets. Every row is another CIFAR10 class as IN-DIST. From left to right: the three images with the lowest anomaly score of the IN-DIST test set, the three images with the lowest anomaly score of the OOD test set and the three images of the IN-DIST test set with the highest anomaly score.

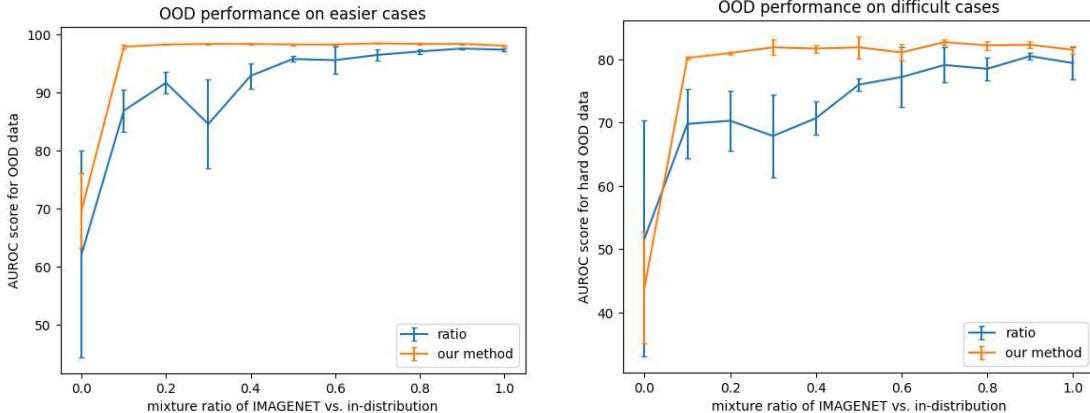

Figure 6: Performance for our model and the ratio method trained on the CIFAR10 horse class as in-distribution and of IMAGENET and in-distribution data as contrastive dataset. On the right we plot the AUROC score against difficult out-of-distribution cases (deer class of CIFAR10), on the left against all other CIFAR10 classes (without horse and deer). Errors are computed over three different seeds.

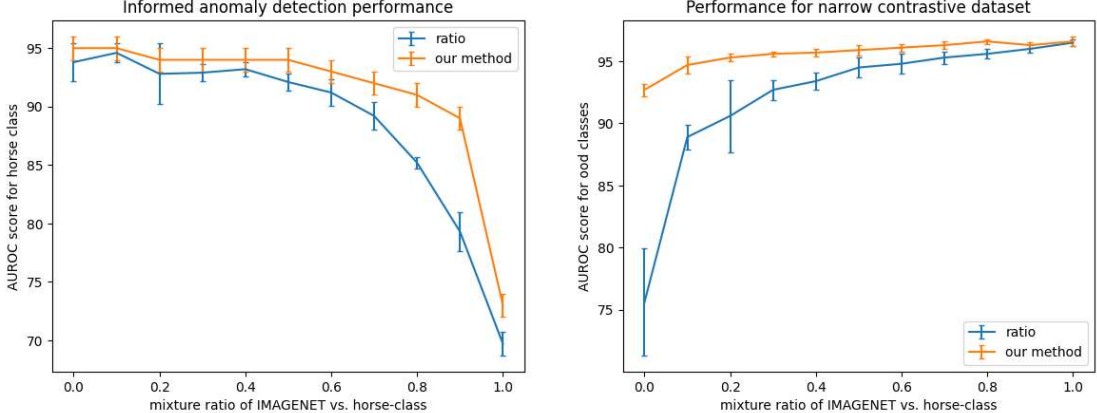

Figure 7: Performance for our model and the ratio method trained on the CIFAR10 deer class as in-distribution and a mixture of IMAGENET and the horse class of CIFAR10 as contrastive dataset. On the left we plot the AUROC score against the horse class included in the mixture (which correspond to a fully supervised setting when the mixture ratio is 0.0), on the right against all other CIFAR10 classes (without horse and deer). For a contrastive class of only horse, images of CIFAR10 (mixture of 0.0), the ratio methods fails, when our method only slightly drops in performance. Standard deviations are computed over three different seeds.

### 5.3.3 Informed anomaly detection

In this ablation study, the performance in a supervised setting, where the contrastive data is a mixture of a broad data distribution and anomalous data is examined. We use the same setting as in the previous experiment: the deer class of CIFAR10 as inlier and a mixture of IMAGENET and the horse class of CIFAR10 as contrastive dataset. We measure the anomaly performance between the inlier class and the horse class - this is in contrast to all other anomaly detection experiments in the paper, as we explicitly use splits of the same distribution as contrastive data and training time and anomalous data at test time. We show the performance for different mixture ratios for our method and a ratio method in figure 7. We see that the performance of our model increases already with a small percentage of anomalous data in the mixture. Both methods saturate at an AUROC score of about 95, we conclude that both classes have a significant overlap in the MOCO feature space and perfect separation is even in the fully supervised case not possible.

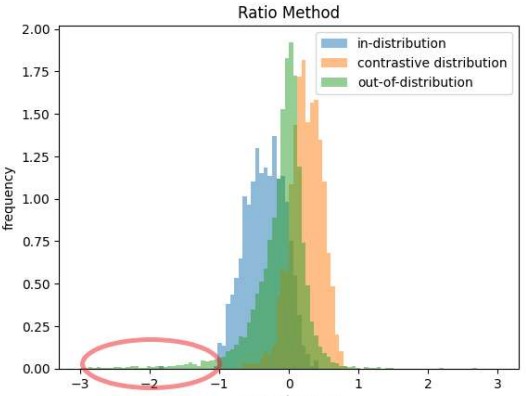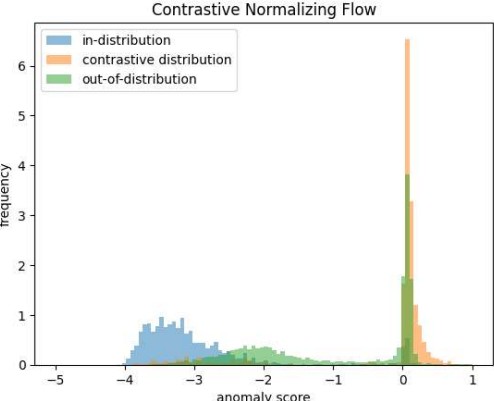

Figure 8: Histograms of the anomaly scores for in-distribution, contrastive and ood data for our model (right) and the ratio method (left) trained on the CIFAR10 deer class as in-distribution and the horse class of CIFAR10 as contrastive dataset. The out-of-distribution data are all other CIFAR10 classes. The models coincide with the models for a mixture ratio of 0.0 in figure 7. While the contrastive flow in-distribution samples have the lowest anomaly score, the tail of the out-of-distribution data (highlighted with a red circle the figure) surpasses the in-distribution data for the ratio method and out-of-distribution samples have the lowest anomaly score.

## 5.4  Beyond the Image Domain

To further investigate our method we apply the contrastive normalizing flow to another data modality: We use the credit card fraud dataset from `https://www.kaggle.com/datasets/mlg-ulb/creditcardfraud`, and apply our contrastive normalizing flow. As a contrastive distribution we use the uncorrelated marginal distributions by permuting all samples in a batch per feature dimension. We show the results in figure 9. The contrastive normalizing flow and the standard normalizing flow perform on par (AUROC of 97.5), where the ratio method fails for anomaly detection (AUROC of 41). Our contrastive normalizing flow does not outperform the standard normalizing flow, which is not surprising: Kirichenko et al. (2020) show that low-dimensional image features dominate the likelihood of the normalizing flow and thus lead to problems on image data. This behaviour is not expected on tabular data, where the standard normalizing flow is already a good anomaly detection method. We think that the contrastive normalizing flow could be useful for other applications in the tabular and other domains: As shown in our ablation study in 5.3.3, the contrastive normalizing flow can be used for anomaly detection with known anomalies.

## 5.5  Discussion and the choice of the contrastive distribution

The experiments show that various contrastive datasets can be used as contrastive distribution and all choices lead to equal or better performance compared to major baselines (standard normalizing flow and likelihood ratio method). As the experiments in subsection 5.3.3 show, the ideal contrastive distribution would consist of out-of-distribution samples only. However, since in most settings these samples are not known at training time, we need to find auxiliary distributions. As we show in the experiments in subsection 5.2.3, a broad auxiliary distribution like IMAGENET works well for our method. This is the same approach taken by Ren et al. (2019). We also show that in contrast to their method, narrow distributions close to the in-distribution data (subsection 5.3.2) and contrastive datasets dissimilar to the test time ood-data (subsection 5.2.3) still lead to superior performance over the ratio method and the standard normalizing flow. We demonstrate that adding inlier data to the contrastive dataset does not lead to a decrease in performance for our method, in contrast to the likelihood ratio approach (section 5.3.1).

For non-image settings, where no broad contrastive dataset like IMAGENET is readily available, the best choice is a set of known outliers. If necessary, this outlier dataset can be enlarged by data augmentation. If no outliers are known at training time, a potential fallback is to construct the contrastive distribution

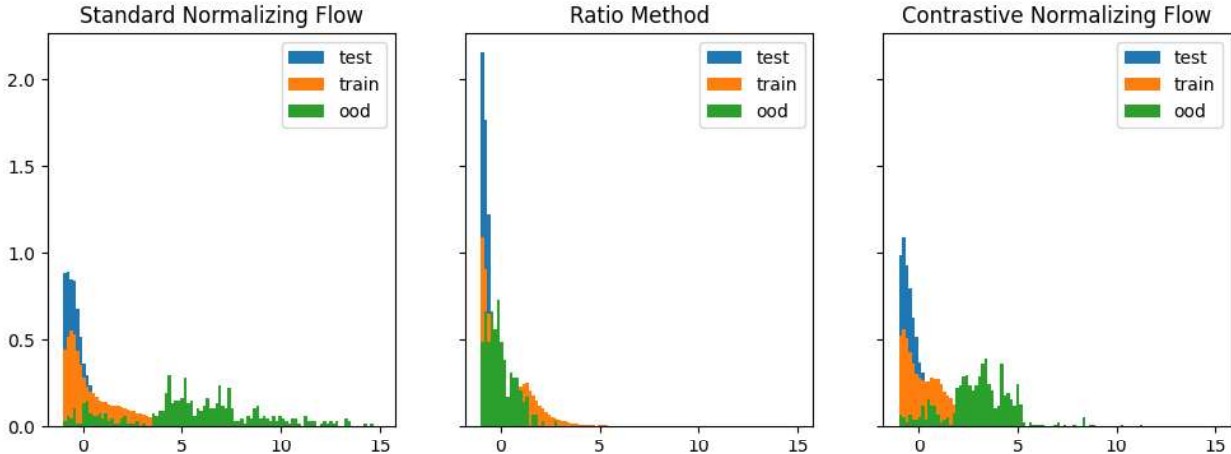

Figure 9: Standard Normalizing Flow, Ratio method and Contrastive Normalizing Flow applied on a tabular fraud detection dataset. While the ratio method completely fails in the tabular domain, the contrastive normalizing flow performs on par with the standard normalizing flow. We think that the contrastive normalizing flow can be used beyond the scope of image anomaly detection in various data domains.

as a transformed inlier distribution, e.g., by adding Gaussian noise or using the product of the marginal distributions removing all correlations. However, we found in the tabular data experiment in section 5.4 that this did not improve performance compared to a standard normalizing flow. We leave the choice of the contrastive distribution in the challenging setting without known outliers or broad auxiliary distribution for future work.

## 6 Conclusion

We propose a novel method to train normalizing flows, which we call contrastive normalizing flow, and show its application to anomaly detection. The method relies on the use of an auxiliary contrastive dataset for training: The training objective is the maximization of the log-likelihood of in-distribution data while minimizing the log-likelihood on the auxiliary dataset. We show that using our training objective, the contrastive normalizing flow learns the normalized positive difference of the in-distribution and the auxiliary distribution. Analysis and the experiments show that the positive difference score can have an advantage compared to the likelihood ratio score when test-outlier data is far away from the contrastive distribution used during training. To increase the computational performance of the method, we employ a pretrained feature extractor on which the contrastive normalizing flow operates. We present advantages and investigate the behaviour of our method on toy and real world experiments, especially compared to the related likelihood ratio approach of Ren et al. (2019). We show that we outperform or perform on par with the benchmark methods on real world data, and do ablation studies to further investigate the influence of the contrastive dataset on the performance. We think that the contrastive normalizing flow can be used for various applications outside of anomaly detection on image data and the scope of this work: It gives the possibility to sample from the - usually intractable - positive difference of distributions.

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
