# OpenReview forum: "Anomaly Detection using Normalizing Flows and Contrastive Data"
_TMLR — Rejected by TMLR_

### Review · Reviewer_wnbF · 2022-10-27

**Summary Of Contributions:**

The paper proposes a contrastive objective for anomaly detection using normalizing flow models.
In particular, the objective maximizes the log likelihood of the inlier distribution while minimizing the log likelihood of auxiliary distribution.
This can be shown to be the positive-difference distribution whose support is only where the inlier distribution is larger than the auxiliary distribution.
In practice, the log likelihood on the auxiliary distribution is clamped to a large negative value during training.
Also, for the sake of experiments, the MOCO contrastive feature extractor was used before performing anomaly detection.
The paper than provides a series of experiments on one-vs-rest outlier detection using classification datasets and dataset-vs-dataset outlier experiments that show comparable or marginal improvements over existing methods such as log likelihood ratio methods.


**Audience:**

Yes

**Broader Impact Concerns:**

No broader impacts section but it does not seem required for this work.


**Claims And Evidence:**

No

**Requested Changes:**

- See experimental weakness for possible updates to the experimental sections.
  - Please move standard deviations to result tables and highlight all methods that are within 1 or 2 standard deviations of the best method. You could also do a statistical test and specify a significance of 0.05 to determine which methods may have the same performance as the "best" method.
  - Please add one experiment using a different feature extractor to determine whether the anomaly method is generalizable beyond MOCO.
  - Please provide hyperparameter tuning specifications for your model as well as baselines.
  - Please add some example ROC curves to inspect.
  - All tables and figures should only be shown *after* they are referenced in the text. It was quite confusing to see various tables before they were introduced. And all baselines should be described before referencing the results.

- The key claim about the method's superiority over log-likelihood ratio should be significantly expanded both in discussion and wth a simplified, targeted experiment(s). Insight into this approach compared to this is the key practical insight.

- (Suggested) The experiments section should be reorganized into a more logical flow of ideas. Right now, it reads more like a random bag of experiments. Rather, perhaps organize each result around a claim that you would like to make and highlight this at the beginning.  Or possibly a question you want to answer with the experiment. These could be smaller and simpler experiments that demonstrate one key thing (e.g., an ablation study).  Then, give the results and discuss whether the hypothesis or question is answered based on the experiments.

- (Suggested) A toy example where your method is definitely better than a log-likelihood ratio approach would be helpful. The main idea of the paper relies on the difference between a log-likelihood ratio vs your positive-difference approach so it is critical to explain and prove this idea at least on a small case where it is easy to understand. Furthermore, at least one experiment should focus on this difference only for real-world datasets.



**Strengths And Weaknesses:**

*Strengths*
- The paper proposes a novel contrastive objective for training a flow model using an auxiliary distribution.

- The paper shows that this objective theoretically learns the positive difference distribution, which is a non-standard distribution that is *proportional to* $(p(x)-q(x))_+ \equiv \max\{0, p(x)-q(x)\}$.

- The experiments include some simple baselines like MSE or ratio-MSE to help give a simple lower bound on performance.

*Weaknesses*
- Experimental results do not support claim of state-of-the-art results. Results are only marginally better than prior works (see e.g., Table 1 and Table 3 where the proposed method is only marginally better on average). It is unclear that the proposed approach supports the claim that the method is state-of-the-art. From the appendix, it seems only 3 runs were done and the standard deviations seem to be around 0.1-1.0 which is definitely within random error for many of the experiments. Thus, it is unclear if these results are statistically significant compared to prior methods.
  - Also, how did you tune the hyperparameters for your method vs the hyperparameters for other methods.  You should tune all methods with roughly the same effort. It is unclear here if or how you tuned the other methods as well.
  - Can you show the ROC curve for each method on at least one experiment?  These curves may reveal significantly more information than the AUROC single number.
  - The experimental setup is fundamentally based on MOCO so it is unclear if the proposed method's better performance is confounded with the choice of MOCO.  Can you do choose one or two other standard latent spaces (e.g., inceptionV3 latent space that is used for FID) to show your method?  Currently, the experimental setup is strongly biased based on MOCO features and thus makes it incomparable to prior work directly or it could be confounded.
  - Another baseline that accounts for samples that are outside of both the inlier and auxiliary distribution is to merely take a new auxiliary distribution that is simply the mixture of the fitted auxiliary density and a Gaussian density where the Gaussian has a small weight like 0.1 or 0.01. Adding this baseline would be helpful in distinguishing the difference between the likelihood ratio approach and the proposed approach.

- The key claim of why this would be better than the log-likelihood ratio approach seems vague. More discussion and more targeted experiments to support this idea should be provided.
  - "An advantage of our method compared to the likelihood ratio is that our learned density is well defined in areas where p and q are very small, i.e,. where the in-distribution data and the contrastive distribution data are sparse." (page 5) This should be explained more with possibly an illustration, a simulated experiment, and a targeted real-world experiment to justify this claim.
  - As another baseline that would tackle this problem, why not just use two outlier detection methods and take an OR operator?  One is for outliers of auxiliary and one is inliers of auxiliary but outliers compared to inlier distribution.

- What is the effect of clamping in Algorithm 1? And perhaps more importantly, how is the threshold chosen?  Log likelihood is different for every dataset---and could be vastly different ranging in orders of magnitude---so choosing a threshold seems very tricky. How was this threshold chosen in the experiments?  How would you suggest choosing this in practice? The effect of this clamping should be more generally discussed, and possibly, an ablation study on different thresholds to show the sensitivity of the method to this clamping value.

- The paper organization and logical flow is problematic and the paper needs a thorough reorganization. For example, the threshold in Alg. 1 was not introduced until the experimental section. See requested changes below about tables/figures.

- What is the effect of adding a small noise to the MOCO features?  Why is this needed? Why aren't the MOCO features already samples from a valid density? It seems they can just be treated as samples from a valid (but unknown) density.

*Minor comments*
- Figure 3 needs to be improved with a nicer looking figure as it is very pixelated and blurry right now.

- Eqs 3- 6 need dx on the outside.  Currently the notation is incorrect.

- Alg 1 should have another loop over the batches or say $\forall i,j$.

---

> ### Author Response · Authors · 2022-11-25
> **Response to Reviewer wnbF (I)**
>
> We want to thank the reviewer for his interesting and in-depth review. We were glad to see the reviewer’s interest and engagement in our novel flow objective and the theoretical derivation. We hope that we could improve the paper through the questions and suggested changes. We added figure 4,6,7,8,9,10,11 and 16 in the revised version and marked changes in the text in blue.
>
> $\textbf{Standard deviations}$
>
> We added the standard deviations to the tables in the main paper and highlighted all methods within one standard deviation from the best performing method. While our method does not clearly outperform all other methods in all experiments, we are convinced that the theoretical and practical benefits (as shown in the new experiments in subsection 5.1 and 5.3) justify our method.
>
> $\textbf{Dependence on MOCO feature extractor}$
>
> We repeated our celebA experiments with inceptionV3 as feature extractor (only the celebA experiment has the minimum required image size for an inception v3 net) and report the results in the appendix. The results are worse than with the MOCO feature extractor (but within two standard deviations). The drop in performance is consistent also for the baselines. We thank you for this important point and hope that this experiment demonstrates that the advantage of our method is not feature space dependent and the MOCO feature extractor is a good choice.
>
> $\textbf{Behavior of the contrastive normalizing flow and the ratio method for distant ood data}$
>
> We added an 2D toy experiment, where our model captures the in-distribution while the ratio method struggles. We also added experiments on the difficult deer/horse class of CIFAR10 to investigate different mixture distribution as contrastive dataset. We show that the difference density learned by our contrastive normalizing flow does not only work well for a broader contrastive dataset but shows also a good performance on a narrow contrastive dataset, such that the OOD data is far from the inlier and the contrastive dataset. We show that the likelihood-ratio does not work well for a narrow contrastive dataset.
>
> $\textbf{Clamping threshold}$
>
> We added two ablation studies regarding the threshold: A simple 1D toy experiment, where we show that the difference-distribution is learned for a wide range of thresholds, with the failure case of a threshold bigger then the in-distribution densities and secondly an ablation on a CIFAR10 class, where the anomaly detection performance stays constant over all chosen thresholds. We added a recommendation how to choose the threshold for an unknown dataset.
>
> $\textbf{Adding noise to the MOCO feature space}$
>
> As described in the experimental section, we normalize the MOCO features, because the MOCO loss only depends on the angle between different feature vectors and not the length. Therefore, the density is only defined on the (d-1) dimensional hypersphere. As all density mass is concentrated at |x| = 1, the density is described by a delta function for the radius. Therefore, it can not be mapped via a finite invertible function to the normal distribution. By adding a small amount of noise, the distribution has full support and can be learned. Additionally, adding a small amount of noise is recommended for training a normalizing flow (https://vislearn.github.io/FrEIA/_build/html/tutorial/tips_tricks_faq.html).
>
> $\textbf{Reorganization of the paper}$
>
> We reorganized the paper and show all figures and tables after or on the same page as their first reference. We added an experimental outline and a small statement before every experiment to motivate it further. We thank you for the minor comments, which we also integrated in our revised version.
>
> $\textbf{Hyperparameter tuning}$
>
> We added a section about our hyperparameter tuning for our method and the baselines in the appendix.
>
> $\textbf{Added gausian to contrastive distribution for ratio method}$
>
>  We did experiments on an analytic mixture of a gaussian and contrastive density on the tabular data, 2D toy example and the experiments in section 5.3 and could not observe an improvement for the likelihood ratio approach and thus do not report the results in the revised version. We hope that the other experiments and ablation studies added in the revised version are helpful in distinguishing the difference between our approach and the ratio method.
>
> $\textbf{Receiver operating characteristic}$
>
> We added ROC curves for the experiment in figure 1 in the appendix in section A.7. We show the curves for different OOD classes for our contrastive normalizing flow, various baseline and ablation methods in figure 16.

---

> > ### Author Response · Authors · 2022-11-25
> > **Response to Reviewer wnbF (II)**
> >
> > $\textbf{Difference between a ratio vs positive-difference approach}$
> >
> > We investigate different mixture distributions as contrastive datasets, e.g., a mixture between IMAGENET, our standard contrastive dataset, and in-distribution. We show that our model is robust to inliers in the contrastive dataset as predicted by our theory, for which we added a small discussion in section 5.3.1, and can be improved by outliers in the contrastive dataset (when having prior knowledge about the anomalies). We compare the performance to the ratio method of Ren et al. (2019) and expand the discussion on theoretical advantages in section 4 and show practical advantages on a toy experiment in section 5.1 and on CIFAR10 classes in section 5.3.

---

> ### Comment · Reviewer_wnbF · 2022-11-30
> **Thank you for the updates**
>
> Thank you for the signficant updates to the paper and the experiments. These help explain the ideas and concepts better. I will take these into consideration in my final recommendation.

---

### Review · Reviewer_8VEy · 2022-10-31

**Summary Of Contributions:**

This work proposes a new perspective to solve anomaly detection by leveraging unlabelled auxiliary datasets and a probabilistic outlier score. The authors use a self-supervised feature extractor trained on the auxiliary dataset and train a normalizing flow on the extracted feature by maximizing the likelihood on in-distribution and minimizing the likelihood on the auxiliary dataset. Multiple benchmark datasets are used to demonstrate the proposed method.

**Audience:**

Yes

**Broader Impact Concerns:**

no concerns

**Claims And Evidence:**

Yes

**Requested Changes:**

I expect to see more discussion about the auxiliary datasets' choice, design, and limitations. Since the work is strongly based on the auxiliary datasets, I suggest to see more comparison and ablation study why the auxiliary plays a critical role on the improvement.

In addition, the title seems a little confusing at the first glance.  Most readers suppose like to combine contrastive learning with normalizing flows so maybe the authors will come up with a better title that matches the core of this work better.

**Strengths And Weaknesses:**

Strength
1. The paper is well-written and organized.
2. The idea is simple and straightforward to follow
3. The work poses a novel direction for AD solutions by leveraging unlabeled existing datasets.

Weakness
1. The novelty is not strong, no major new contributions from the methodology side
2. The auxiliary datasets are selected for specific tasks or can be any related datasets
3. The choice of auxiliary datasets is somehow difficulty in real-world applications

---

> ### Author Response · Authors · 2022-11-25
> **Response to Reviewer 8VEy**
>
> We thank the reviewer for his review. We are glad that the reviewer liked our intuitive and straightforward approach and found the paper well written and organized. We added figure 4,6,7,8,9,10,11 and 16 in the revised version of the paper and marked changes in the text in blue.
>
> $\textbf{Discussion about the auxiliary datasets' choice, design, and limitations}$
>
> We agree that the choice of the dataset is fundamental to our method and therefore added new experiments to the revised version of our paper: We investigate different mixture distributions as contrastive datasets, e.g., a mixture between IMAGENET, our standard contrastive dataset, and in-distribution. We show that our model is robust to inliers in the contrastive dataset as predicted by our theory and can be improved by outliers in the contrastive dataset (when having prior knowledge about the anomalies). We compare the performance to the ratio method of Ren et al. (2019) and show theoretical and practical advantages, e.g., that our model still performs well for inlier contamination of the contrastive dataset. We show that the ratio method does not have the same advantages as our method for these contrastive distributions and that the advantages shown in the 2D toy experiment also hold on real world data.
>
> $\textbf{Novelty}$
>
> We admit that the idea is simple in nature and easy to implement but see this as a strength. Despite its simplicity, the method remedies theoretical and practical problems of the existing likelihood ratio method as we show in the 2D toy example in subsection 5.1 and the ablation studies in 5.3.
>
> $\textbf{Title might be confusing}\$
>
> We see that the title might be confusing to some readers and changed it to “Anomaly Detection using Normalizing Flows and Contrastive Data”.

---

### Review · Reviewer_9EAt · 2022-11-11

**Summary Of Contributions:**

This paper claims to propose a new approach to anomaly detection. The problem setting is image specific. The underlying assumption seems that there is a background auxiliary distribution q(x) that represents anomalous samples. Thus, the goal is to learn a generative model $p_\theta$ so that the "contrast" to q(x) is maximized.

The authors claim that they leverage the notion of normalizing flow, which seems to be just an external black-box model to provide a probability distribution for x.

**Audience:**

No

**Broader Impact Concerns:**

The proposed method seems very image specific. Application to other domains will not be straightforward.

**Claims And Evidence:**

No

**Requested Changes:**

- Define the task. Explicitly explain the task. What is the proposed anomaly score?
- Clearly articulate the novelty. The role of the aux distribution and how general such an assumption is are not very clear.

**Strengths And Weaknesses:**

Strengths
- develops a new objective to learn a probability distribution.

Weakness
- The framework seems to contain three technical components, which are MOCO embedding, the normalizing flow model, and the "contrastive" objective. However, they look at individually picked building blocks that lack a coherent narrative.
- The problem of anomaly detection is not clearly defined. The authors seem to make a quite specific assumption on what should be detected as anomalous, but the text lacks a clear-cut explanation.

---

> ### Author Response · Authors · 2022-11-25
> **Response to Reviewer 9EAt**
>
> We thank the reviewer for his review. We added figure 4,6,7,8,9,10,11 and 16 in the revised version of the paper and marked changes in the text in blue.
>
> $\textbf{Task}$
>
> We clarify our used anomaly score in the revised version of the paper: We use the negative log-likelihood of the difference distribution described in the paper, which is given by the output of our normalizing flow. We also added a clear definition of our semantic anomaly detection task in the introduction. We want to detect images which stem from another semantic distribution as the learned inlier class (e.g., learn on the deer class of CIFAR10 and detect all other CIFAR10 classes as anomalies), as is common in the literature (see  Bergman and Hoshen 2020, Tack et al.
> 2020, Hendrycks et al. 2019)
>
> $\textbf{Novelty and Normalizing Flow}$
>
> We reformulated parts of our introduction to point out our novelty, which is the use of an additional dataset for normalizing flow training and the theoretical derivation showing that our method learns a usual intractable difference density (the normalized positive difference between in-distribution and contrastive dataset). While the former is also possible for other kinds of generative models, the theoretical background is special to normalizing flows. Normalizing flows are the only deep generative model with exact normalized likelihoods which can model arbitrary target distributions.
>
> $\textbf{Building blocks}$
>
> We clarified in the revised version of the paper that the feature extractor is an optional component which is used in the image domain for computational performance reasons. For the tabular and toy data, we do not use a feature extractor.
>
> $\textbf{Role of contrastive distribution}$
>
> We added experiments to show the role and influence of the contrastive distribution: We investigate different mixture distributions as contrastive datasets, e.g., a mixture between IMAGENET, our standard contrastive dataset, and in-distribution. We show that our model is robust to inliers in the contrastive dataset as predicted by our theory and can be improved by outliers in the contrastive dataset (when having prior knowledge about the anomalies). We compare the performance to the ratio method of Ren et al. (2019) and show theoretical and practical advantages, e.g., that our model still performs well for inlier contamination of the contrastive dataset. We show that the ratio method does not have the same advantages as our method for these contrastive distributions and that the advantages shown in the 2D toy experiment also hold on real world data.
>
> $\textbf{Image specific}$
>
> We think that the application of contrastive normalizing flows on datasets beyond the image domain (e.g., tabular data or time series) is an interesting research question: We added a small experiment on tabular data in subsection 5.4. Normalizing flows on tabular data does not suffer the same problems for anomaly detection as in the image domain, e.g., that local correlations dominate the likelihood. Therefore, the Standard Normalizing flow works well on tabular data for anomaly detection without our extension and is not outperformed by our model. Nonetheless, we think that there exist interesting applications and advantages of our method for non-image data, e.g., sampling from the usual intractable difference distribution.
>
> $\textbf{References}$:
>
> Bergman and Hoshen. Classification-based anomaly detection for general data. 2020.
>
> Jihoon Tack, Sangwoo Mo, Jongheon Jeong, and Jinwoo Shin. CSI: novelty detection via contrastive learning on distributionally shifted instances. NeurIPS, 2020.
>
> Dan Hendrycks, Mantas Mazeika, Saurav Kadavath, and Dawn Song. Using self-supervised learning can improve model robustness and uncertainty. NeurIPS, 2019

---

> > ### Comment · Reviewer_9EAt · 2022-11-28
> > **Thank you for updating the manuscript**
> >
> > Thank you for updating the manuscript and I apologize for not responding in a timely manner.
> >
> > The new version looks more understandable than the original. However, I still do not think the paper clearly and properly frames the problem. One of my main concerns was that the task was stated in an image-specific way and lacked a clear-cut definition. The updated version still seems to share that issue. For example, the auxiliary distribution does not seem to be clearly defined in the paper. It was introduced in the context of image classification. Many readers will be left confused, wondering what a counterpart in non-image use cases would be.
> >
> >
> > From a methodological point of view, technical contributions still do not seem to be clearly explained.
> >
> > The use of the negative log-likelihood as the anomaly score is a standard method. The use of distributional divergence is a common approach in change/anomaly detection, too. We need more clarity regarding the novelty in light of existing works.
> >
> > The authors highlight their contribution in replacing the distributional ratio with the distributional difference in the updated version. However, it is not clear how it advanced the existing divergence-based anomaly detection frameworks. There have been many attempts in anomaly detection using various statistical distances, such as the Hellinger distance, the total variation, or their variants. Again, we need more clarity in light of the existing approaches.
> >
> > The normalizing flow approach is one way of doing density estimation, if I understand correctly. On the other hand, the discussion of Section 4, p.5, does not seem to be related with the specific approach of normalizing flow. $p_\theta$ could be estimated with any method, like VAE or even Gaussian mixtures. If the authors wish to claim their contribution in developing a new version of the normalizing flow approach, it should be clearly explained in that section. This is one of the things I meant by lacking a coherent narrative.

---

> > > ### Author Response · Authors · 2022-11-29
> > > **Thank you for your response**
> > >
> > > Thanks for your comment and the helpful clarification of the original review. We updated our submission again according to your questions. The changes compared to the last update are highlighted in light blue.
> > >
> > > $\textbf{Choice of contrastive distribution and non-image domain}$
> > >
> > > In the new subsection 5.4 (which we added after your initial review), we conduct an experiment on tabular data. We now added another new subsection 5.5 in the experimental part where we discuss all experiments and give recommendations on the choice of the contrastive distribution. There, we also acknowledge that for non-image data the choice of the contrastive distribution may be difficult, when no potential outliers are available for use as contrastive dataset. We leave this as a topic for future research.
> > >
> > >
> > > $\textbf{Technical Novelty}$
> > >
> > > While it is common to define outliers via the negative log-likelihood under the *data distribution*, we believe that an outlier score using NLL under the proposed *difference distribution* is novel. This idea leads to a new training objective and theory and subsequently to our new anomaly score.
> > >
> > > $\textbf{Hellinger distance, TV-distance and other distributional distances}$
> > >
> > > We added a paragraph to the introduction:
> > >
> > > >"In contrast to distribution-based anomaly detection, where a set of many test-samples/measurements is compared to the training distribution (see e.g. Coluccia et al. (2013); Bouyeddou et al. (2018)) we focus on the single-shot setting, where the inlier vs. outlier decision must be made independently for individual data points."
> > >
> > > This setting is also used by Ren et al. (2019) and Schirrmeister et al. (2020), and we see no strong relation to the distribution-based setting, where e.g. Hellinger distance or TV-distance are applicable. However, we would be very interested in additional references you might provide shedding more light on the relationship between these settings.
> > >
> > > $\textbf{Importance of Normalizing Flow}$
> > >
> > > We agree that our new training objective can be used with other density estimators and added a clarification to the introduction:
> > >
> > > >"We focus on normalizing flows as density estimators, since they are universal approximators working well for high dimensional data.
> > > However, the new objective and theory is applicable to any density estimator trainable with explicit maximum likelihood, as is required for our new objective in equation (3).
> > > This includes Mixture-of-Gaussian density estimators or binning methods in lower dimensional settings."
> > >
> > >
> > > Reference:
> > >
> > > Shyamal Dhua, Akash Bhosale, Nikhil Chaudhari, Saurabh Dhapre. Anomaly Detection Using Hellinger Distance. 2013.
> > >
> > > Jie Ren, Peter J. Liu, Emily Fertig, Jasper Snoek, Ryan Poplin, Mark A. DePristo, Joshua V. Dillon, and Balaji Lakshminarayanan. Likelihood ratios for out-of-distribution detection. 2019.
> > >
> > > Robin Schirrmeister, Yuxuan Zhou, Tonio Ball, and Dan Zhang. Understanding anomaly detection with deep invertible networks through hierarchies of distributions and features. 2020.
> > >
> > > Angelo Coluccia, Alessandro D’Alconzo, and Fabio Ricciato. Distribution-Based Anomaly Detection in Network Traffic. 2013.
> > >
> > > Benamar Bouyeddou, Fouzi Harrou, Ying Sun, and Benamar Kadri. An effective network intrusion detection using hellinger distance-based monitoring mechanism. 2018.

---

### Decision · Action_Editors · 2022-12-23

**Recommendation:** Reject

**Comment:**

As discussed, the reviewers have expressed concerns regarding some of the paper's claims and the paper's organization and clarity. The authors made a notable effort to address the reviewers' points in the revised version, adding experimental results and extra discussion. However, some concerns remained in the reviewers' final recommendations. In light of this, it is clear that the paper would benefit from a thorough revision and a second round of reviewing. Therefore, I recommend rejection at this stage, but I would like to encourage the authors to consider resubmitting a revised version to TMLR.

As a minor editorial point, I would encourage the authors to improve the visual quality of the figures, as the current figures use low-quality graphics. In particular, I would recommend using vector graphics wherever possible, especially for figures that display plots and text.


**Audience:**

The paper studies an important problem (anomaly detection) using advanced machine-learning techniques (normalizing flows, contrastive learning, self-supervised feature extractors). Therefore, the paper is clearly of interest to TMLR's audience.

That said, 2 out of 3 reviewers expressed concerns about the paper's organization and clarity, which could potentially hamper the paper's reach to TMLR's audience. Specifically:
- Reviewer wnbF stated that the paper's organization and logical flow is problematic, and pointed to potential improvements.
- Reviewer 9EAt stated that the task is not clearly defined, that the ideas lack a coherent narrative, and that the exposition is overly image-specific.


**Claims And Evidence:**

In their final recommendations, 2 out of 3 reviewers stated that the paper's claims are not fully supported by evidence.

In particular, one of the claims of the paper is that the proposed method is state-of-the-art for anomaly detection. Reviewer wnbF stated that it is unclear whether the experimental results support this claim fully, as the improvement over the baselines is marginal and not convincingly statistically significant.

In light of the reviews, it seems that the paper should either revise the claim about the method being state-of-the-art, or provide stronger empirical evidence to support the claim.